# Tournament Style RL: Stabilizing Policy Optimization on Non Verifiable Problems

**Gurusha Juneja** [1][†]  **Shubham Milind Phal** [2]  **Jennifer She** [2]  **Lisa Wang** [2]  **Dorsa Sadigh** [2][3]  **Anca Dragan** [2][4]
**William Yang Wang** [1]

## Abstract

Many real-world tasks are non-verifiable—there is no objective ground truth, and quality must be judged subjectively—making reward design for RL difficult. Existing approaches based on scalar rubric scores or single comparisons are often noisy, poorly calibrated, or provide sparse learning signals. We introduce Tournament Style RL (`TSRL`), which constructs rewards from rubric-guided pairwise judgments against a fixed set of anchor responses, using win-rate as the reward for policy optimization. This aggregation of comparisons against anchor responses yields a signal that is more robust to the judge noise by stabilizing the reference frame, reducing the variance in reward. We test across four non-verifiable tasks and two backbone LLMs, and find that `TSRL` improves average win-rate by $+43.8$ points over the base model and $+22.8$ points over the strongest baseline. `TSRL` scales with the number of anchors, remains robust under weak or partially corrupted judges, the results are supported by blinded human preference studies.

## 1. Introduction

Many important real-world problems are non-verifiable, i.e. there is no objective ground truth or deterministic notion of correctness. This includes mental health counseling support, educational tutoring and feedback, policy and safety advice, negotiation, decision support, etc where the quality is inherently subjective. Unlike semi-verifiable tasks, these problems admit no automatic evaluation signal, and even human assessments expressed as absolute scores (e.g., 0–10)

vary significantly across annotators. Reinforcement learning has achieved strong results in domains with objectively verifiable rewards, such as games and code generation (Sutton et al., 1998; Mnih et al., 2013; Li et al., 2022). However, scaling RL to non-verifiable problems remains unsolved. This is primarily because the absence of a ground truth makes reliable reward design difficult (Ziegler et al., 2019; Liu et al., 2025).

Prior work has explored several approaches for learning in non-verifiable domains. Preference-based methods such as RLHF (Christiano et al., 2017; Ouyang et al., 2022; Touvron et al., 2023) rely on human comparisons, but they are costly to scale and provide very sparse and noisy feedback (Zheng et al., 2023). Rubric-based scoring (Liu et al., 2025; Jia et al., 2025), typically using LLM judges, assigns a scalar score to responses based on a predefined set of rubrics. Such scores are coarse: two responses that satisfy the same number of rubrics but differ meaningfully in quality can receive identical rewards. A third direction uses model-internal signals, such as self-confidence (Zhao et al., 2025) or predictive entropy (Zhang et al., 2025; Agarwal et al., 2025) as rewards, but this can reinforce hallucinated or confidently wrong behavior.

To address these limitations, we introduce **Tournament Style RL (`TSRL`)**.[1] It replaces brittle scalar rewards with a stable, ordinal learning signal. We develop on the key insight, that, relative pairwise judgments against a fixed set of anchor responses provide a more robust signal than scalar rubric scores or single comparisons. This is because they (a.) replace an ill-calibrated absolute scale with a shared reference, and (b.) aggregate multiple pairwise outcomes which provides robustness to individual mis-orderings and reduction in variance of the reward signal.

**Example.** To further explain point (b), assume each pairwise decision is independently flipped with probability $e$, and we compare a response against $m$ anchors, defining the reward $R$ as its win-rate. Then $\mathrm{Var}(R) = \frac{e(1-e)}{m}$, while for a single comparison $S \in \{0, 1\}$, $\mathrm{Var}(S) = e(1 - e)$. Thus,

---

[†]Work done during an internship at Google DeepMind. [1]University of California, Santa Barbara [2]Google DeepMind [3]Stanford University [4]University of California, Berkeley. Correspondence to: Gurusha Juneja <gurusha@ucsb.edu>, Shubham Milind Phal <shubhamphal@google.com>.

*Proceedings of the 43rd International Conference on Machine Learning*, Seoul, South Korea. PMLR 306, 2026. Copyright 2026 by the author(s).

[1]Our code is available at https://github.com/UCSB-AI/tsrl.

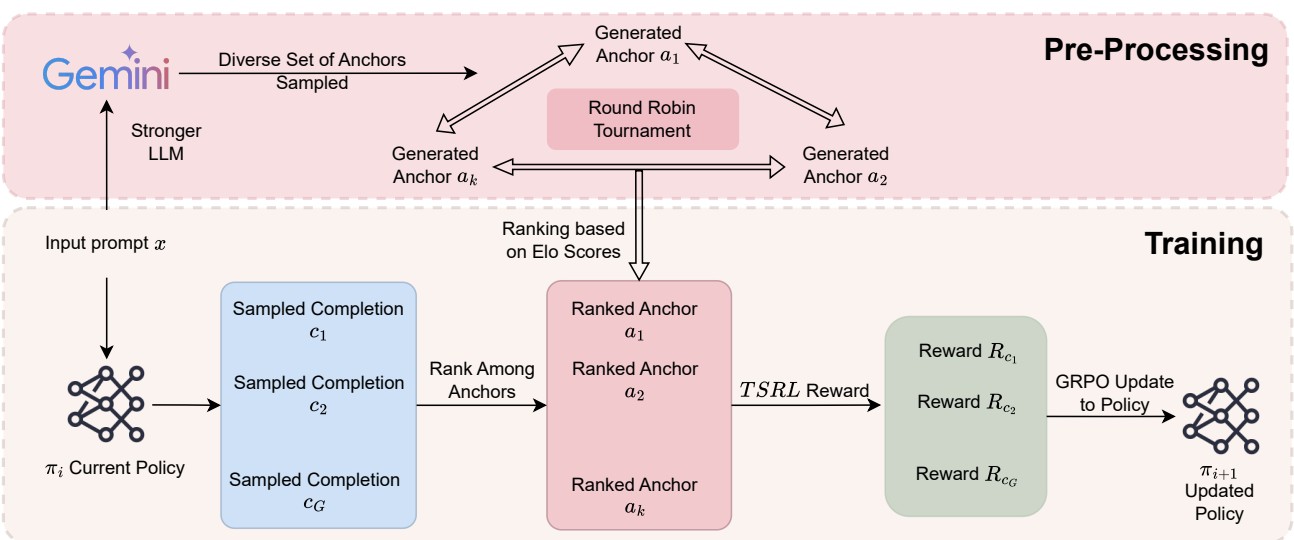

*Figure 1.* **Tournament Style RL (TSRL).** Pre-processing: for each prompt $x$, we sample a diverse anchor set $\mathcal{A} = \{a_1, \ldots, a_k\}$ from a stronger LLM and run a round-robin tournament to produce a ranked anchor ladder (e.g., via Elo-style aggregation). Training: the current policy $\pi_i$ samples a group of completions $\{c_1, \ldots, c_G\}$; each $c_j$ is compared against every anchor $a_\ell$ using a rubric-guided LLM judge (criteria first, quality as tie-break), yielding outcomes $\mathbb{I}[c_j \succ a_\ell]$. TSRL assigns each completion a win-rate reward $R(c_j) = \sum_{\ell=1}^{k} \mathbb{I}[c_j \succ a_\ell]/k$, and uses these rewards in a GRPO update to obtain $\pi_{i+1}$. Fixed anchors provide a stable reference frame, while aggregating many comparisons yields a reward robust to judge noise and reduced variance than scalar rubric scores or single-pair feedback.

using $m = 8$ anchors yields an $8\times$ reduction in variance compared to a single comparison. This reduction in noise provides a much stronger signal to the model and leads to sample efficient learning.

TSRL maintains a small set of anchor responses per prompt by sampling from a strong model. It then ranks them via a round-robin tournament, evaluating on task-specific rubric items and overall quality. During training, each response sampled form the current policy is compared against all anchors under the same rubric-guided evaluation, and rewarded with the resulting win-rate.

We evaluate TSRL on four non-verifiable tasks—story generation, constrained humor generation, webpage design generation, and mental health counseling. Across both backbones (LLaMA-3.1-8B and Qwen-3-4B), TSRL improves over the base model by $\approx 43.8$ points on average and over the strongest baseline (rubric-based RL) by $\approx 22.8$ points. Beyond empirical gains, we analyze tournament rewards under a noisy-judge comparator model and show that aggregating comparisons against a fixed anchor set reduces mis-ordering probability and lowers reward variance, making the relative advantages used by GRPO more reliable. Consistent with this, TSRL scales favorably with the number of anchors and degrades gracefully under weak or partially corrupted judges. Human preference studies confirm that these improvements align with human judgments.

**Contributions.**

1. **Tournament reward for non-verifiable RL.** We introduce TSRL, to train models on non-verifiable tasks by converting rubric-guided pairwise judgments into a win-rate against a fixed set of anchor responses, rather than optimizing noisy scalar rubric scores or sparse single comparisons.

2. **Theoretical analysis on robustness and stability of TSRL.** We prove that aggregating multiple pairwise outcomes yields lower-variance reward estimates, under noisy judge, than single comparisons, which in turn implies more stable gradients during policy optimization (under standard independence assumptions).

3. **Empirical validation.** Across all four tasks, TSRL consistently outperforms baselines, achieving **+33.5** and **+32.0** absolute win-rate gains on stories and humor over the strongest baseline, while remaining robust to weaker or corrupted judges; human evaluations corroborate these improvements.

## 2. RL for Non-Verifiable Problems

In this section we introduce TSRL, and provide theoretical analysis of noise robustness and convergence stability compared to rubric based and single comparison based training regimes.

## 2.1. Problem Setting and Notation

Given a prompt $x$ (e.g. a story writing prompt/topic) and a policy $\pi_\theta(\cdot \mid x)$, let $c \sim \pi_\theta(\cdot \mid x)$, $c \in \mathcal{C}$, be the output response, where $\mathcal{C}$ is the space of possible responses for the input $x$. Because the task is non-verifiable, there is no single ground-truth correct response. Instead, each response has an approximate latent quality score, $q : \mathcal{C} \to \mathbb{R}$. We want to train $\pi_\theta$ to produce responses with higher latent $q$.

Since, $q(c)$ is unobserved, past works either optimize $\pi_\theta$ based on comparison between two sampled responses (Xu et al., 2020; Wang et al., 2025) or by estimating $q$ by scoring against a rubric, using LLM Judge (Bhat, 2023; Hashemi et al., 2024). We present a method of optimizing $\pi_\theta$ that provides a richer signal that is more robust to the judge's noise.

As many past works do (Christiano et al., 2017), we assume access to a pairwise LLM based (possibly noisy) Judge that acts like a comparator. Given two responses $a, b$, it provides us with $I[a \succ b] \in \{0, 1\}$ with probability $\Pr(a \succ b) = f(q(a) - q(b))$, where $f : \mathbb{R} \to (0, 1)$ *is a strictly increasing bounded function* (Bradley & Terry, 1952; Luce et al., 1959; Shepard, 1957).

## 2.2. Tournament Style Reinforcement Learning

In TSRL, we first generate $k$ fixed anchor responses denoted as $\mathcal{A} = \{r_1, \ldots, r_k\}$ for each prompt $x$. We then compare anchors pairwise via the judge, construct a directed comparison graph, and perform a topological sort to impose a global ordering over anchors. During training, for a sampled candidate response $c \sim \pi_\theta(\cdot|x)$, we compare $c$ against each anchor $r_i$, obtaining binary outcomes $I[c \succ r_i]$. We define the win-rate reward:

$$w(c) = \frac{\sum_{i=1}^{k} I[c \succ r_i]}{k}$$

We use $w(c)$ as the reward signal in a GRPO update, i.e.:

$$\nabla_\theta J \approx \mathbb{E}_{c \sim \pi_\theta}\big(w(c) - b\big) \nabla_\theta \log \pi_\theta(c),$$

where $b$ is the group mean.

**Comparator and noise model.** We assume a (possibly noisy) pairwise judge such that for any anchor $r_i \in \mathcal{A} = \{r_1, \ldots, r_k\}$,

$$I[c \succ r_i] \sim \text{Bernoulli}(p_i(c)), \qquad p_i(c) = f(q(c) - q(r_i))$$

We further assume conditional independence across anchors given $(c, \mathcal{A})$.[2] For rubric scalar scoring baselines, we model $s(c) = q(c) + \varepsilon$ with $\varepsilon \sim \mathcal{N}(0, \sigma_{\text{rub}}^2)$ i.i.d. across candidates.

---

[2]This is the standard simplifying assumption in pairwise-comparison analyses, which holds unless there is a systematic bias in the comparator model.

**Monotonicity of the expected win-count.** Define the tournament win-rate reward $w(c) = \sum_{i=1}^{k} I[c \succ r_i]/k$ and its expectation $\mu(c) = \mathbb{E}[w(c)] = \sum_{i=1}^{k} p_i(c)/k$. Since $f$ is strictly increasing, for any $c_1, c_2$ with $q(c_1) > q(c_2)$ we have $p_i(c_1) > p_i(c_2)$ for all $i$, hence

$$\Delta\mu \triangleq \mu(c_1) - \mu(c_2) = \sum_{i=1}^{k} \big(p_i(c_1) - p_i(c_2)\big)/k > 0.$$

This means higher latent quality implies higher expected win-count against a fixed anchor set. Therefore, any observed reversal $w(c_2) \geq w(c_1)$ is attributable to stochastic judge noise rather than systematic bias.

## 2.3. TSRL provides better noise robustness

TSRL assigns each sampled response $c \sim \pi_\theta(\cdot|x)$ a reward $w(c)$ computed only via comparisons to a fixed anchor set. In GRPO, however, updates depend on relative advantages within a group (e.g., $w(c) - b$ where $b$ is the group mean). If the reward is noisy, a lower-quality sample can receive higher reward than a higher-quality one, flipping the sign of the advantage and injecting noise into the policy gradient. We therefore analyze the probability that the tournament reward misorders two candidates $c_1, c_2$ with $q(c_1) > q(c_2)$.

**1. Misordering probability under TSRL.** Let $D \triangleq w(c_1) - w(c_2) = \sum_{i=1}^{k} Z_i/k$, where $Z_i \triangleq I[c_1 \succ r_i] - I[c_2 \succ r_i] \in [-1, 1]$ and $\mathbb{E}[D] = \Delta\mu$. By Hoeffding's inequality (using independence across $i$),

$$\Pr\big(w(c_2) \geq w(c_1)\big) = \Pr(D \leq 0)$$
$$= \Pr(D - \Delta\mu \leq -\Delta\mu) \leq \exp\Big(-\frac{(\Delta\mu)^2 k}{2}\Big) \quad (1)$$

if we assume "well-spread anchors" i.e. there exists $\delta > 0$ and $\alpha \in (0, 1]$ such that for at least $\alpha k$ anchors, $p_i(c_1) - p_i(c_2) \geq \delta$. then $\Delta\mu \geq \alpha\delta$, and (1) yields

$$\Pr\big(w(c_2) \geq w(c_1)\big) \leq \exp\big(-ck\big) \quad \text{for } c = \frac{\alpha^2\delta^2}{2}.$$

Hence, as the number of anchors grows, the probability that noisy rewards flip the ordering between a better and worse candidate decays exponentially in $k$ under this model.

**2. Single-pair comparison.** If rewards are defined using a single anchor $r$ as $S(c) = I[c \succ r]$, then for $c_1, c_2$ the event $S(c_2) > S(c_1)$ occurs only when $c_2$ wins and $c_1$ loses, so

$$\Pr\big(S(c_2) > S(c_1)\big) = p_2(1 - p_1), \quad p_j = f(q(c_j) - q(r)).$$

Unlike the tournament case, this does not improve with $k$; its value depends entirely on the (possibly small) margin to the chosen anchor.

**3. Rubric scoring.** For the scalar baseline $s(c) = q(c) + \varepsilon$ with $\varepsilon \sim \mathcal{N}(0, \sigma_{\text{rub}}^2)$ i.i.d.,

$$\Pr\big(s(c_1) < s(c_2)\big) = \Pr\big(\varepsilon_1 - \varepsilon_2 < -(q(c_1) - q(c_2))\big)$$
$$= \Phi\Big(-\frac{q(c_1) - q(c_2)}{\sqrt{2}\,\sigma_{\text{rub}}}\Big).$$

This misordering probability decreases with increase in quality gap but is constant and cannot be further reduced for a given $c_1, c_2$.

---

**Takeaway.** Under standard independence assumptions, `TSRL` converts noisy pairwise decisions into an aggregated win-count whose ordering error shrinks rapidly with the number of anchors. Since GRPO updates depend on relative advantages across samples, reducing reward misorderings directly improves the reliability of the gradient signal used for policy optimization.

---

### 2.4. `TSRL` yields lower-variance rewards

In the previous subsection we analyzed misordering probability Here we study a complementary notion of stability: the variance of the reward signal itself. Intuitively, a reward with lower variance produces less noisy advantages, which reduces stochasticity in policy-gradient updates.

**1. Tournament reward (anchor win-rate).** Since, we defined the reward as the average of $k$ independent Bernoulli comparisons, by independence,

$$\mathbb{E}[w(c)] = \frac{1}{k} \sum_{i=1}^{k} p_i(c),$$

$$\text{Var}(w(c)) = \frac{1}{k^2} \sum_{i=1}^{k} p_i(c)\big(1 - p_i(c)\big) \ \leq \ \frac{1}{4k}.$$

Hence, for fixed candidate $c$, the variance of the tournament reward decreases at least as fast as $1/k$. Equivalently, each additional anchor comparison contributes another bounded term to an average, and the law of large numbers reduces reward noise through aggregation.

**2. Single-pair reward (like in RLHF).** The reward is a single Bernoulli outcome against one anchor $r_j$. Then

$$\mathbb{E}\big[I[c \succ r_j]\big] = p_j(c),$$

$$\text{Var}(I[c \succ r_j]) = p_j(c)\big(1 - p_j(c)\big) \ \leq \ \frac{1}{4}.$$

Thus, the intrinsic reward noise is $O(1)$ and does not shrink unless we increase supervision (e.g., repeat comparisons and average). Note that here we assume that the anchor is fixed, but in RLHF, the anchor itself is variable, inducing extra noise.

**3. Rubric scalar reward (single noisy score).** The reward is defined as a single scalar rubric score $s(c) = q(c) + \varepsilon$, $\varepsilon \sim \mathcal{N}(0, \sigma_{\text{rub}}^2)$. For a fixed candidate $c$,

$$\mathbb{E}[s(c)] = q(c), \qquad \text{Var}(s(c)) = \sigma_{\text{rub}}^2.$$

Unlike the tournament reward, this variance does not systematically decrease with additional structure unless one explicitly averages multiple independent rubric evaluations.

**Implication for gradient stability in GRPO.** GRPO updates use centered rewards (advantages). The gradient estimator is given by:

$$\hat{g}(c) = \big(R(c) - b\big)\nabla_\theta \log \pi_\theta(c \mid x),$$

where $b$ is typically the within-group mean reward. When $b$ is estimated from a finite group, the variance of the centered advantage $R(c) - b$ is proportional to $\text{Var}(R(c))$ up to a factor $(1 - 1/G)$ (for group size $G$) (Wasserman, 2004), so lower reward variance directly reduces advantage noise. Under a standard bounded-score-function assumption (or weak dependence between $R$ and $\|\nabla \log \pi\|$), this reduces the variance of the policy-gradient estimator and yields smoother optimization dynamics (Chung et al., 2021; Mao et al., 2018). To summarize, tournament win-rate rewards shrink reward variance as $1/k$, while single-pair rewards remain $O(1)$ and rubric scoring retains an irreducible $\sigma_{\text{rub}}^2$ per evaluation.

---

**Takeaway.** In GRPO, learning is driven by relative advantages within a sampled group. Tournament win-rate rewards make these advantages more reliable by reducing variance (random fluctuations) in the reward assigned to each sample. As a result, GRPO updates are less sensitive to individual judge errors, which empirically corresponds to sample efficiency and smoother training curves.

---

## 3. Experimental Setup

**Tasks and Datasets.** We evaluate Tournament Style Reinforcement Learning (`TSRL`) on four non-verifiable generative tasks: story generation, constrained humor generation, webpage design generation, and mental health counseling. These tasks admit multiple plausible outputs and require subjective quality assessment, making verifiable reward design extremely difficult.

For the **story generation** task, we use Writing Prompts dataset (Fan et al., 2018), which contains short narrative prompts paired with approximately ten human-written story completions per prompt. We use the story narrative prompts as inputs $(x)$ and treat the human-written completions as the anchor responses.

For the **Constrained humor generation** task, we use derived from the MWAHAHA (Castro et al., 2025) shared task

*Table 1.* We report win-rate (%) against the reference set of anchors (Higher is better) on held-out test set of prompts for four domains: story generation, constrained humor generation, webpage design generation, and mental health counseling. Results are shown for two backbones (LLaMA-3.1-8B and Qwen-3-4B). TSRL achieves the best performance across all tasks and both models, and the ablations highlight key design factors: performance degrades under a weaker judge (+ Bad Judge), but the degradation is not much and collapses when anchors lack diversity (+ Low Diversity), indicating that reliable evaluation and diverse anchors are critical to tournament-based training.

| | **LLaMA-3.1-8B** | | | | | **Qwen-3-4B** | | | |
|---|---|---|---|---|---|---|---|---|---|
| **Method** | **Stories** | **Humor** | **Web** | **MentalH.** | **Method** | **Stories** | **Humor** | **Web** | **MentalH.** |
| Base Model | 13.5 (1.8) | 54.0 (2.1) | 40.5 (1.4) | 30.2 (1.9) | Base Model | 12.8 (1.6) | 52.4 (2.3) | 41.2 (1.9) | 30.5 (1.4) |
| SFT | 14.0 (1.5) | 56.0 (1.2) | 47.2 (2.3) | 34.8 (1.7) | SFT | 14.5 (1.2) | 55.2 (1.8) | 48.0 (1.5) | 35.1 (2.2) |
| One Anchor | 13.5 (2.4) | 60.0 (1.9) | 42.1 (1.6) | 32.5 (2.0) | One Anchor | 13.2 (2.1) | 58.6 (1.4) | 43.5 (2.0) | 32.8 (1.7) |
| RLHF (RM) | 40.5 (1.1) | 58.0 (2.2) | 54.0 (1.5) | 44.7 (1.8) | RLHF (RM) | 39.2 (1.9) | 57.5 (2.5) | 55.4 (1.3) | 45.2 (2.1) |
| Rubrics | 47.5 (1.3) | 62.0 (1.6) | 61.2 (2.4) | 50.9 (1.2) | Rubrics | 46.1 (1.4) | 61.2 (1.1) | 62.8 (2.2) | 51.7 (1.8) |
| Intuitor | 14.2 (2.0) | 56.4 (1.7) | 46.8 (2.1) | 34.1 (1.5) | Intuitor | 14.8 (2.3) | 54.9 (1.6) | 47.5 (1.9) | 34.6 (2.0) |
| **TSRL** | **74.0** (0.4) | **94.0** (0.3) | **76.8** (0.5) | **67.2** (0.4) | **TSRL** | **73.2** (0.5) | **93.1** (0.2) | **78.5** (0.4) | **68.4** (0.5) |
| + Bad Judge | 70.0 (0.6) | 91.0 (0.5) | 71.2 (0.7) | 61.5 (0.6) | + Bad Judge | 68.5 (0.7) | 89.2 (0.6) | 72.1 (0.8) | 62.3 (0.7) |
| + Low Diversity | 12.8 (1.5) | 55.4 (1.4) | 39.1 (2.6) | 30.8 (1.5) | + Low Diversity | 12.1 (2.4) | 54.3 (1.5) | 40.8 (0.6) | 31.2 (1.4) |

in which the model is given two target words and is required to generate a short joke that incorporates both words. Success in this task depends on semantic relevance, coherence, and humor quality, all of which are highly subjective. Since we did not have human written responses for this, we use Gemini-2.5-Pro (Comanici et al., 2025) to generate 10 jokes per word-pair conditioning on a spectrum of quality.

In the **Webpage design generation** task, we use the Web-SightDescribed (Laurençon et al., 2024) dataset. The datset contains natural language description of the webpage along with a reference html to generate that webpage. The task requires the model to produce HTML code given a natural language description of a webpage. The generated HTML is rendered and provided to the evaluator (Kimi-K2 VL (Team et al., 2025)), which judges the visual layout, structural correctness, adherence to the specification, and finally outputs a winner based on the overall quality of the rendered image. Although the structural correctness of html code can be verified, the image quality is subjective. . This task emphasizes multimodal reasoning and design quality beyond syntactic correctness.

For the **Mental health counseling** task, we use MentalChat16K (Xu et al., 2025) dataset. The dataset consists of an open-ended user prompt $(x)$ requiring emotional assistance and a human counselor's response. The model is required to generate an appropriate response to the given user prompt. Each query can be answered in different ways and one problem could have multiple valid solutions. Based on human counselor's responses, we construct rubrics for each sample and judge it for helpfulness.

**Rubric-Based Judges and Anchor Construction.** For each task, we design a task-specific rubric that decomposes quality into multiple interpretable criteria (e.g., relevance, coherence, creativity, constraint satisfaction). The rubric

prompts used by the judge are provided in Appendix E. we use Kimi-K2 (Team et al., 2025) as the judge (unless otherwise stated). Using these rubrics, we perform round robin tournaments among a fixed set of diverse anchor responses, sampled from Gemini-2.5-Pro or taken from human responses depending on availability for, each prompt. Based on the outcomes of these comparisons, we provide a score to each response, which induces a ranking over the anchor responses, this serves as a stable reference pool throughout training.

During training, candidate model completions are compared pairwise against each anchor response using the same rubric-guided judge. The judge calculates each completion's score based on the rubrics and returns a binary decision, as to which response is better, based on the rubrics and overall quality of the completion. Aggregating these outcomes yields a tournament-style win-rate reward, which reflects the relative quality of the candidate with respect to the anchor pool.

**Evaluation Protocol.** Fore each task, we split the data into disjoint train and held-out test set. Each data-point includes the input $(x)$ paired with ranked anchor completions. For each test prompt, we sample responses from the trained model and compute their relative rank using the same rubric-based judge employed during training. Performance is reported as win-rate, defined as the fraction of test instances where the model's sampled completion outperforms the reference completions. Each experiment is run 3 times and the mean is reported.

**Models and Training.** We conduct experiments using two base language models: **LLaMA 3.1 8B** and **Qwen 3 4B**. Both models are trained using Group Relative Policy Optimization (GRPO) (Shao et al., 2024). For each training

prompt, we sample a group of $N = 8$ candidate responses from the current policy. For each response, we compute the tournament-style reward by comparing it against the anchor set as described above. GRPO then normalizes rewards within the group and updates the policy accordingly.

All methods are trained under identical optimization settings, rollout budgets, and sampling procedures to ensure fair comparison. Additional implementation details are provided in Appendix B.

**Baselines.** We compare TSRL against a diverse set of baselines representing common approaches to learning in non-verifiable domains. These include: (i) **Base Model**, which is the win-rate of the base model, (ii) **Supervised Fine-Tuning (SFT)/ Distillation**, we finetune the model on the top half of the anchor responses which are sampled using Gemini-2.5-pro; (iii) **Rubric-based reward optimization**, where a single scalar reward is derived from scores that the judge gives to the completion based on the same rubrics used in TSRL judge. (Gunjal et al., 2025); (iv) **Single-pair based methods**, like RLHF (Christiano et al., 2017; Ouyang et al., 2022), which train policies using direct comparisons between candidate responses without reference anchors. We do this in two ways, first, following Ouyang et al. (Ouyang et al., 2022), we train a reward model (LLaMA-3.1-8B) using the anchor responses as pairs, where the reward model outputs 1 if the first response is better (**RLHF (RM)**). Second, we use an LLM-Judge to do a single pairwise comparison with the best anchor (**One Anchor**). and (v) **Intrinsic Rewards (Intuitor)**, which uses intrinsic signals from the model, like entropy (Zhao et al., 2025), as a reward to train on non-verifiable tasks.

# 4. Results

## 4.1. Comparison with Existing Techniques

We compare TSRL against the representative set of baselines described in Sec 3 (see Table 1). For this experiment, we use $k = 4$ anchors, since we see diminishing returns post that (discussed in Sec 4.2). TSRL issues $k$ judge calls per candidate response while the single-comparison baselines issue one, so we run the other baselines (One Anchor, RLHF, Rubrics, and Intuitor) for $4\times$ the gradient steps of TSRL to equalize the total number of judge calls across methods. Under this equal judge-budget protocol TSRL still reaches a higher final win-rate, which shows that the gains come from the stable reference frame rather than from additional judge supervision. We analyze the compute and judge-cost tradeoff in Sec 4.11. We find that TSRL improves the win-rate over the baselines across all tasks and models. It achieves an absolute average gain of **40.0** over SFT and **22.6** over rubric-based methods on LLaMA-3.1-8B, and **40.1** over SFT and **22.9** over rubric-based methods on Qwen-3-

4B.

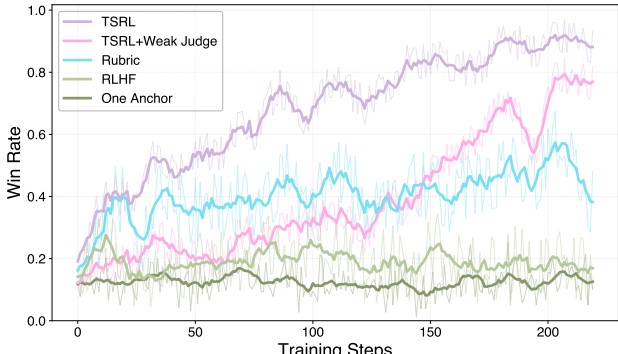

*Figure 2.* Mean win-rate against four anchor responses on the validation set over training steps for TSRL and the baselines. TSRL achieves higher rewards with lower variance than baselines, converging in fewer steps. The sample efficiency reflects the reduced variance and a more consistent training signal. We only show till step 220 here for brevity but other baselines are evaluated at the best step up till step 880 to account for additional judge calls made by TSRL.

We further find that TSRL exhibits improved training dynamics compared to the baselines. As can be seen in Figure 2, the mean reward increases smoothly for TSRL and TSRL+weak judge compared to other baselines. Furthermore, we see that it takes less train steps to reach the same reward, this shows that TSRL is sample efficient, which is a direct result of low variance in rewards. These results support our claims in Sec 2.4 about variance reduction and training stability using TSRL.

## 4.2. Scaling the Number of Anchors

Next, we investigate how performance scales with the number of anchors used to construct the tournament reward. We vary the number of anchors $k \in \{1, 2, 4, 5, 6, 8, 10\}$ while keeping all other components fixed, including the training data, number of updates, and optimization hyper-parameters.

Figure 3 shows that increasing the number of anchors consistently improves final performance. TSRL yeilds substantial performance gains from $k = 1$ (equivalent to a single pairwise comparison) to $k = 4$. But the rate of increase in perfomrance decreases post $k = 4$ anchors, making it the optimal number of anshors given the added inference cost. Across tasks, performance improves by an average of **52 win-rate points** from $k = 1$ to $k = 10$.

## 4.3. Effect of Anchor Diversity

Finally, we examine how the diversity of the anchor pool affects learning. We construct two types of anchor sets: first, **high-diversity** (original), where the anchors span varied styles and strategies. This is constructed specifically

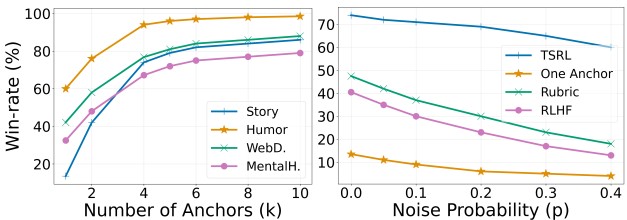

*Figure 3.* Effect of number of anchors in `TSRL` and it's robustness to judge noise. **[Left]** The plot shows the win rate of the final checkpoint trained using `TSRL` with $k$ anchors. We find increasing the number of anchors improves performance with gains saturating beyond $k \approx 4$. **[Right]** `TSRL` degrades gracefully under increasing noise, substantially outperforming rubric based and single-comparision based baselines.

by prompting the stronger LLM (Gemini-2.5-Pro) with different prompts. Second, **low-diversity** anchors consisting of similar responses. `TSRL` is trained with identical $k$ and compute across both settings. As shown in Table 1, models trained with high-diversity anchors consistently outperform those trained with low-diversity anchors by **43.6** win-rate points on held-out data. This indicates that anchor diversity is an important design factor in `TSRL`.

### 4.4. Effect of Judge Noise

To study `TSRL`'s robustness to evaluator quality, we train the using a significantly weaker judge LLM (LLaMA-3.1-8B instead of Kimi-K2), while keeping anchors, prompts, and training hyperparameters unchanged. Comparing to the strong-judge setting (Table 1), `TSRL` retains approximately **94.6%** of its performance under the weak judge (an average drop of 4.2 points across Stories/Humor/MentalH.).

For a fair comparison, we also train the other baselines with the same weak judge (Table 2). Rubric-based training is substantially more sensitive to judge quality, dropping by **11.2** points on average across the three tasks (e.g., **13.5** points on Stories). Single-pair (one-anchor) training also degrades under the weak judge—most notably on Humor (**8** points)—but remains far below `TSRL` in absolute performance. These results indicate that `TSRL` offers better robustness against judge noise, validating our claim in Sec 2.3.

### 4.5. Artificial Noise Injection

We further do a controlled experiment to probe `TSRL`'s robustness against the judge noise. We artificially inject noise into the judge's decisions by independently flipping each pairwise judgment with probability $p \in \{0, 0.05, 0.1, 0.2, 0.3, 0.4\}$, while keeping all other training parameters the same.

Figure 3 plots final performance as a function of $p$. Performance for all methods degrades as noise increases. How-

*Table 2.* Performance under a weak or miscalibrated judge for LLaMA-3.1-8B. We report win-rate (%) against human reference responses. TSRL degrades gracefully under evaluator noise, while scalar-reward and single-pair methods suffer substantial performance drops.

| Method (Weak Judge) | Stories | Humor | MentalH. |
|---|---|---|---|
| Base Model | 13.5 | 54.0 | 30.2 |
| One Anchor | 12.0 | 52.0 | 30.5 |
| Rubric-based Judge | 34.0 | 51.0 | 41.8 |
| **TSRL** | **70.0** | **91.0** | **61.5** |

ever, for `TSRL`, the performance degradation is significantly slower. At $p = 0.3$, TSRL retains over **80%** of its clean-judge performance, whereas single-pair and scalar-reward methods collapse sharply, losing over **40%** of their gains.

### 4.6. Generalization across Model Scales

The gains of `TSRL` hold across model sizes. On story generation, Qwen2.5-3B-Instruct improves from 10.0 (base) and 45.0 (rubric) to **59.2**, and Qwen2.5-7B-Instruct from 15.3 and 50.0 to **78.0**. The gap over the rubric baseline grows from **14.2** to **28.0** points with scale, indicating that a stronger policy benefits more from a stable reference frame.

### 4.7. Comparison with Preference Optimization

An alternative to the on-policy win-rate reward is to optimize the same pairwise preferences offline with DPO (Rafailov et al., 2023). Training LLaMA-3.1-8B with DPO on preference pairs built from the anchor responses reaches **57.5** win-rate on story generation, against **74.0** for `TSRL`. On-policy optimization against a fixed reference ladder gives a stronger signal than offline preference optimization over the same comparisons.

### 4.8. Effect of Anchor Quality

Beyond diversity, anchor quality affects training. We construct story-generation anchors from human writers, Gemini-2.5-Pro, and the weaker Gemini-2.5-Flash, and train LLaMA-3.1-8B with each (Table 3). Higher-quality anchors give higher win-rate, from **65.5** (Flash) to **70.0** (Pro) to **74.0** (human), all well above the rubric baseline (47.5). A check with GPT-5.4 as judge confirms the ordering: Pro anchors beat Flash anchors in **81.2%** of comparisons, and the Pro-anchored model beats the Flash-anchored model in **54.5%** of cases.

### 4.9. Disentangling the Stable Reference Frame

To separate the stable reference frame from the variance reduction of multiple comparisons, we add a baseline that ranks the $G = 8$ in-batch completions against each other and

*Table 3.* Win-rate (%) on story generation (LLaMA-3.1-8B) by anchor source. Higher-quality anchors yield higher performance, and all sources beat the rubric baseline.

| Method | Win Rate |
|---|---|
| Base model | 13.5 |
| Rubrics (strongest baseline) | 47.5 |
| TSRL (Gemini-2.5-Flash anchors) | 65.5 |
| TSRL (Gemini-2.5-Pro anchors) | 70.0 |
| **TSRL (human anchors)** | **74.0** |

rewards the in-batch rank. On story generation (LLaMA-3.1-8B), in-batch ranking reaches **55.0** win-rate against **74.0** for TSRL with $k = 8$ fixed anchors. The fixed reference frame is the main driver, since in-batch ranking still optimizes against a moving baseline.

### 4.10. Self-Bootstrapped TSRL without an External Model

TSRL sources anchors from a stronger model or human writers, so the reward depends on an external supervisor. We test a variant that removes this dependence: we sample the anchor set from the policy being trained using four quality-tier prompts (low-effort, mediocre, skilled, top-tier), and refresh the anchors from the current policy every 50 steps. We train LLaMA-3.1-8B on story generation under the main hyperparameters (Appendix B) and evaluate against the same held-out human anchors.

Table 4 shows self-bootstrapped TSRL reaches **81.5** win-rate, **8.5** points above standard TSRL (74.0) and **34.0** above the rubric baseline, with no external model. Refreshing anchors from the improving policy acts as a curriculum that compounds with the stable reference frame, giving a preliminary route to a ranking reward without an external supervisor.

### 4.11. Compute and Judge-Cost Analysis

TSRL issues $k$ judge calls per candidate against one for the single-comparison baselines, so for batch size $B$, group size $G$, and $T$ steps it makes $B\,G\,T\,k$ judge calls against $B\,G\,T$. To compare under a fixed judge budget, we run the baselines for $4\times$ the gradient steps of TSRL ($k = 4$), giving comparable judge spend (about $21.12 training and $3.96 inference per run). Even then, TSRL at 220 steps (**74.0**) exceeds the rubric baseline (**47.5**) and RLHF (RM) (**40.5**) at 880 steps, and trains in 2.5 hours against roughly 7 for the rubric baseline. TSRL is therefore more efficient per unit of judge compute, not merely per gradient step.

### 4.12. Human Study

To verify that TSRL actually aligns with human preferences and the LLM-Judge's alignment with human responses, we

*Table 4.* Self-bootstrapped TSRL on story generation (LLaMA-3.1-8B). Anchors are re-sampled from the training policy every 50 steps via four quality-tier prompts, with no external model. Win-rate is measured against the same held-out human anchors as the main experiments.

| Method | Step | Win Rate | $\Delta$ |
|---|---|---|---|
| Base model | 0 | 13.5 | — |
| Self-Bootstrap (Iter 1) | 50 | 40.5 | +27.0 |
| Self-Bootstrap (Iter 2) | 100 | 77.5 | +64.0 |
| Self-Bootstrap (Iter 3) | 150 | 80.0 | +67.5 |
| Self-Bootstrap (Iter 4) | 200 | **81.5** | +69.0 |
| TSRL (external anchors) | 220 | 74.0 | +60.5 |
| Rubrics | 880 | 47.5 | +34.0 |

run a blinded human evaluation on the held-out test set of prompts. We focus on two non-verifiable tasks: *constrained joke generation* and *story generation*. We evaluate only *LLaMA-3.1-8B*'s generations.

**Setup.** For each task, we sample 50 held-out input prompts. For every input, we collect generations from our method and from all baselines, and additionally include the anchor responses used by TSRL. Annotators are shown the prompt and a brief task description (e.g., required constraints for jokes, and general quality criteria for stories), and then asked to rank all candidates from best to worst. Each input is labeled by 6 independent annotators. The study is blinded: model/baseline identities are hidden and candidates are randomly ordered (See Appendix D).

**Agreement.** To assess label reliability, we compute inter-annotator agreement across the full ranking data. We observe substantial agreement: **0.554** on jokes and **0.606** on stories. This is substantially high given the nature of the task (ranking). We also find a strong spearman rank correlation ($\rho$ **0.833**, $p$ **0.01**) between the LLM Judge rankings and human rankings.

**Metrics.** From the ranked lists, we compute human win-rates by counting how often one method is ranked above another across all annotators and inputs. We report win-rate comparisons of TSRL against key baselines and references.

**Results and alignment with the judge.** Human rankings strongly favor TSRL. Averaged across all comparisons against the baselines in the pool, TSRL achieves win-rates of **82.92** on jokes and **84.17** on the stories task. Against the base model, TSRL attains **68.3** on jokes and **73.3** win-rate on stories; against the rubric-based method, **53.3** and **56.7**; and against RLHF, **53.3, 68.3** on the jokes and stories task respectively. These results are consistent with trends measured by the LLM judge, indicating that the tournament-based reward improvements reflect human-perceived quality gains rather than judge-specific artifacts.

## 5. Related Work

**RL with Verifiable Rewards.** Most RL successes rely on verifiable rewards computed from an objective ground truth, enabling direct supervision (Sutton et al., 1998; Mnih et al., 2013). In LLMs, verifiable rewards appear in domains with well-defined correctness criteria (e.g., math, code, symbolic tasks) and have supported large-scale RL training (Lewkowycz et al., 2022; Li et al., 2022; Wei et al., 2022). Group-based policy optimization methods (e.g., GRPO-style updates) further stabilize training by normalizing feedback within sampled candidate sets (Shao et al., 2024). In contrast, our focus is on non-verifiable tasks where there is no deterministic correctness signal. Rather than relying on sparse signals, TSRL constructs an ordinal reward from repeated relative judgments against a fixed reference set, allowing more stable and reliable optimization compared to the previous methods.

**RL for Non Verifiable Problems: preferences, rubrics, and internal signals.** A broad line of work replaces ground-truth rewards with subjective feedback, most commonly via preference-based RL (PbRL) and RLHF, which learn from human or model-based pairwise comparisons (Christiano et al., 2017; Ouyang et al., 2022; Touvron et al., 2023). While effective, pairwise feedback is low-bandwidth and often used either through two-stage pipelines (reward modeling then RL) or through within-batch comparisons that (a) amplify any noise in the judge, and (b) inherit a moving reference frame as the policy improves. In parallel, rubric-based methods use structured criteria and LLM judges to assign rewards or scores (Liu et al., 2025; Jia et al., 2025; Gunjal et al., 2025), but when compressed into a single scalar these rewards can be poorly calibrated and too coarse to distinguish responses that satisfy the same rubric items yet differ in quality. Other approaches attempt to bypass external evaluators by using model-internal signals such as confidence, self-consistency, or predictive entropy as rewards (Zhao et al., 2025; Zhang et al., 2025; Agarwal et al., 2025), but these can reinforce confidently wrong behavior in non-verifiable domains. Finally, LLM-as-judge and dueling-style methods embed comparisons directly into optimization, reducing reliance on explicit reward models but typically comparing only contemporaneous samples (Zheng et al., 2023). TSRL differs from these approaches by (i) evaluating each candidate against a fixed anchor pool that defines a stable reference frame, and (ii) aggregating many rubric-guided comparisons into a win-rate reward, increasing feedback bandwidth and reducing sensitivity to individual judge errors.

**Ranking Models and Tournament Aggregation.** Statistical ranking models such as Bradley–Terry / Elo and Plackett–Luce formalize how latent utilities can be inferred from noisy comparisons (Bradley & Terry, 1952; Luce et al., 1959; Berg, 2020; Plackett, 1975), and spectral or graph-based methods provide additional guarantees for recovering orderings under comparison noise (Negahban et al., 2018). Related ideas appear in learning-to-rank and dueling bandits, where repeated comparisons and aggregation reduce inconsistency and improve selection (Wu & Liu, 2016; Yue et al., 2012). TSRL is inspired by this literature but uses it for a different goal: we do not fit a global ranking model via maximum likelihood; instead, we use a fixed anchor ladder to define a simple win-rate reward for RL. This "anchored tournament" construction yields a reward whose variance decreases with the number of anchors when normalized as win-rate, and (under standard independence assumptions) reduces the probability that noise reverses relative orderings—properties that directly improve the reliability of GRPO-style advantage estimates.

## 6. Conclusion and Future Work

**Conclusion.** We studied reinforcement learning for non-verifiable tasks, where no objective ground truth exists and reward design must rely on subjective, noisy evaluators. We introduced **Tournament Style RL (TSRL)**, which replaces brittle scalar rewards and sparse single comparisons with an ordinal win-rate signal from rubric-guided pairwise judgments against a fixed set of anchor responses. Across four non-verifiable domains and two backbone models, TSRL improves win-rate over strong baselines, scales with the number of anchors, and stays robust under weak or corrupted judges, with gains confirmed by blinded human studies.

**Future work.** A key open question is how TSRL behaves as the policy improves beyond the quality range covered by the anchor set. If the policy becomes consistently stronger than the anchors, comparisons saturate (win-rates approach 1), reducing reward variance and weakening the learning signal. One natural extension is to refresh anchors over training—periodically re-sampling a diverse anchor pool from the improved policy (or a mixture of policy snapshots) to maintain a discriminative reference ladder. This would turn TSRL into a self-improving procedure that reduces reliance on external anchor sources, and could ultimately enable training from model-generated reference sets once a sufficiently strong initial policy is available. We report a preliminary result in this direction in Sec 4.10, where anchors are sampled from the training policy and refreshed every 50 steps, matching and then exceeding TSRL with external anchors and removing the dependence on a stronger model. More broadly, designing principled anchor-refresh schedules, diversity constraints, and safeguards against drift or collapse remains an important direction for scalable RL in subjective domains.

# Impact Statement

This paper presents work whose goal is to advance the field of Machine Learning. There are many potential societal consequences of our work, none which we feel must be specifically highlighted here.

**Use of Large Language Models.** In compliance with the ICML guidelines, we used large language models only to refine prose and improve grammatical clarity in parts of the paper. Concretely, we provided hand-written bullet-point notes and asked the model to polish the writing. All core ideas, experimental designs, research artifacts, and scientific content, including all equations, were developed entirely by the human authors.

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

# A. Proofs

**1. Under Tournament Reward** $w(\cdot)$**:**   We are interested in the probability $\Pr\big(w(c_2) \geq w(c_1)\big)$, which denotes the event where the lower-quality candidate appears to win more or an equal number of comparisons. Let $X_i = I[c_1 \succ r_i]$ and $Y_i = I[c_2 \succ r_i]$. These are independent Bernoulli random variables with probabilities $p_i(c_1)$ and $p_i(c_2)$ respectively. The win-counts are $w(c_1) = \sum_{i=1}^{k} X_i$ and $w(c_2) = \sum_{i=1}^{k} Y_i$. Define a new random variable $Z_i = X_i - Y_i$. Note that $Z_i$ can take values $\{-1, 0, 1\}$. The difference in win-counts is $W = w(c_1) - w(c_2) = \sum_{i=1}^{k}(X_i - Y_i) = \sum_{i=1}^{k} Z_i$.

The expectation of $W$ is $\mathbb{E}[W] = \sum_{i=1}^{k} \mathbb{E}[X_i - Y_i] = \sum_{i=1}^{k}(p_i(c_1) - p_i(c_2)) = \mu_1 - \mu_2 = \Delta\mu$. The variance of $W$ is $\mathrm{Var}(W) = \sum_{i=1}^{k} \mathrm{Var}(X_i - Y_i)$ due to the independence of comparisons across different anchors. Since $X_i$ and $Y_i$ (for the \*same\* anchor $r_i$) are independent Bernoulli trials given the candidates, $\mathrm{Var}(X_i - Y_i) = \mathrm{Var}(X_i) + \mathrm{Var}(Y_i)$. Thus, $\sigma^2 = \mathrm{Var}(W) = \sum_{i=1}^{k}[p_i(c_1)(1 - p_i(c_1)) + p_i(c_2)(1 - p_i(c_2))]$.

We want to bound $\Pr\big(w(c_2) \geq w(c_1)\big) = \Pr\big(W \leq 0\big)$. Since $\mathbb{E}[W] = \Delta\mu > 0$, we are bounding the probability that $W$ falls below its mean. For sums of independent random variables, a Chernoff-type bound or a direct application of the concentration inequality relating to the standard deviation yields:

$$\Pr\big(W \leq 0\big) \leq \exp\!\Big(-\frac{(\Delta\mu)^2}{2\,\sigma^2}\Big).$$

As $k$ grows, if anchors are well spread, each term $p_i(c_1) - p_i(c_2)$ contributes positively to $\Delta\mu$, leading to $\Delta\mu = O(k)$. Similarly, each variance term $p_i(c)(1 - p_i(c))$ contributes to $\sigma^2$, making $\sigma^2 = O(k)$. Thus, the exponent $\frac{(\Delta\mu)^2}{2\,\sigma^2}$ scales as $\frac{O(k^2)}{O(k)} = O(k)$. Consequently, the misordering probability decays exponentially with $k$, approximately $\exp(-ck)$ for some constant $c > 0$.

**Under Single-Pair Comparison to the same anchor** $r$**:**   Let $p_1 = \Pr(c_1 \succ r) = f(q(c_1) - q(r))$ and $p_2 = \Pr(c_2 \succ r) = f(q(c_2) - q(r))$. Since $q(c_1) > q(c_2)$ and $f$ is strictly increasing, $p_1 > p_2$. Misordering occurs if $I[c_2 \succ r] = 1$ and $I[c_1 \succ r] = 0$. Given independence of comparisons $I[c_1 \succ r]$ and $I[c_2 \succ r]$:

$$\Pr\big(I[c_2 \succ r] = 1 \text{ and } I[c_1 \succ r] = 0\big) =$$

$$\Pr(I[c_2 \succ r] = 1) \cdot \Pr(I[c_1 \succ r] = 0) = p_2 \cdot (1 - p_1).$$

This probability is a fixed value and does not decay with $k$.

**3. Under Rubric Scalar Scoring:**   We have $s(c_1) = q(c_1) + \varepsilon_1$ and $s(c_2) = q(c_2) + \varepsilon_2$, where $\varepsilon_1, \varepsilon_2 \sim \mathcal{N}(0, \sigma_{\mathrm{rub}}^2)$ are independent. Misordering occurs if $s(c_1) < s(c_2)$, which implies $q(c_1) + \varepsilon_1 < q(c_2) + \varepsilon_2$, or $\varepsilon_1 - \varepsilon_2 < q(c_2) - q(c_1)$. Let $\Delta q = q(c_1) - q(c_2)$. We know $\Delta q > 0$. So the condition is $\varepsilon_1 - \varepsilon_2 < -\Delta q$. The difference $Z = \varepsilon_1 - \varepsilon_2$ follows a normal distribution: $Z \sim \mathcal{N}(0, \sigma_{\mathrm{rub}}^2 + \sigma_{\mathrm{rub}}^2) = \mathcal{N}(0, 2\sigma_{\mathrm{rub}}^2)$. Therefore, the misordering probability is:

$$\Pr\big(s(c_1) < s(c_2)\big) = \Pr(Z < -\Delta q) = \Phi\left(-\frac{\Delta q}{\sqrt{2}\,\sigma_{\mathrm{rub}}}\right).$$

This probability is also a constant, dependent on $\Delta q$ and $\sigma_{\mathrm{rub}}$, and does not decay with $k$.

## B. Training Details

### Model.

- Base model: Meta-Llama-3.1-8B-Instruct and Qwen-3-4B-Instruct-2507
- Full fine-tuning (no LoRA)
- Max prompt/response length: 512 tokens

### Optimization.

- Optimizer: AdamW
- Learning rate: $5 \times 10^{-7}$
- Weight decay: 0.01
- Gradient clipping: 1.0

### Algorithm.

- RL algorithm: GRPO
- KL coefficient: 0.001
- Rollout samples: 16

### Training.

- Batch size: 8
- Epochs: 15
- Hardware: $8\times$ A100 GPUs
- Training time: 23–90 hours (method-dependent)

**Policy optimizer.**    `TSRL` defines a reward and is independent of the policy optimizer. We use GRPO (Shao et al., 2024) for all results in the main paper, but the same win-rate reward applies to any on-policy method, including PPO (Schulman et al., 2017) and REINFORCE. The noise-robustness and variance-reduction analysis in Sec 2.4 relies on the standard REINFORCE gradient estimator and therefore does not depend on the choice of optimizer.

## C. Extending `TSRL` to Partially-Verifiable Tasks

`TSRL` targets non-verifiable tasks, but the win-rate reward also applies to partially-verifiable ones where solutions can be ordered by degree of correctness. On MATH (Hendrycks et al., 2021), we use four anchors ranging from a fully incorrect solution to one with an early mistake, a late mistake, and the gold solution. Training LLaMA-3.1-8B for 100 steps with Kimi-K2 as judge, `TSRL` reaches **54.5** accuracy against **50.0** for a binary final-correctness reward and **47.0** for the base model. Ordering by partial correctness gives a denser signal than a binary outcome reward.

# D. Human Study

## Joke Ranking Annotation Tool
### Task Instructions

**What you'll be doing:** You will be helping us evaluate AI-generated jokes by ranking them based on how funny they are. Each annotation task shows you two words (e.g., "carousel" and "lemonade") and 12 different jokes that incorporate both words.
**Your task:** Rank these 12 jokes from funniest to least funny (or least sensible) based on your personal judgment. Drag and drop jokes to reorder them, with the funniest at the top (position 1) and least funny at the bottom (position 12).
**Time estimate:** Approximately 2-3 minutes per joke pair.
**Important:** Your data is stored only in your browser, not on the server. Click "Download CSV" regularly to save your progress. When you finish your assigned range, download the final CSV file.

☐ **I have read and understood the instructions**

*Figure 4.* Instructions screen for the human annotation interface. Annotators read the task description and confirm understanding before proceeding.

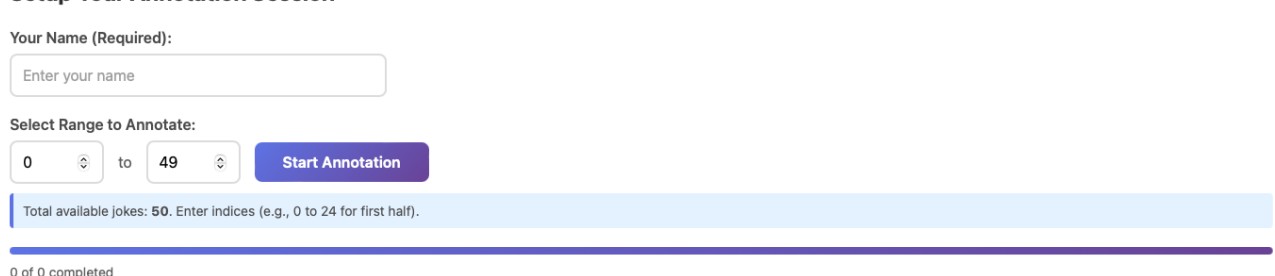

## Setup Your Annotation Session

**Your Name (Required):**

Enter your name

**Select Range to Annotate:**

0  ⇅   to   49  ⇅   **Start Annotation**

Total available jokes: **50**. Enter indices (e.g., 0 to 24 for first half).

0 of 0 completed

*Figure 5.* Session setup screen where annotators enter their name and select a range of samples to annotate. Progress is tracked locally in the browser.

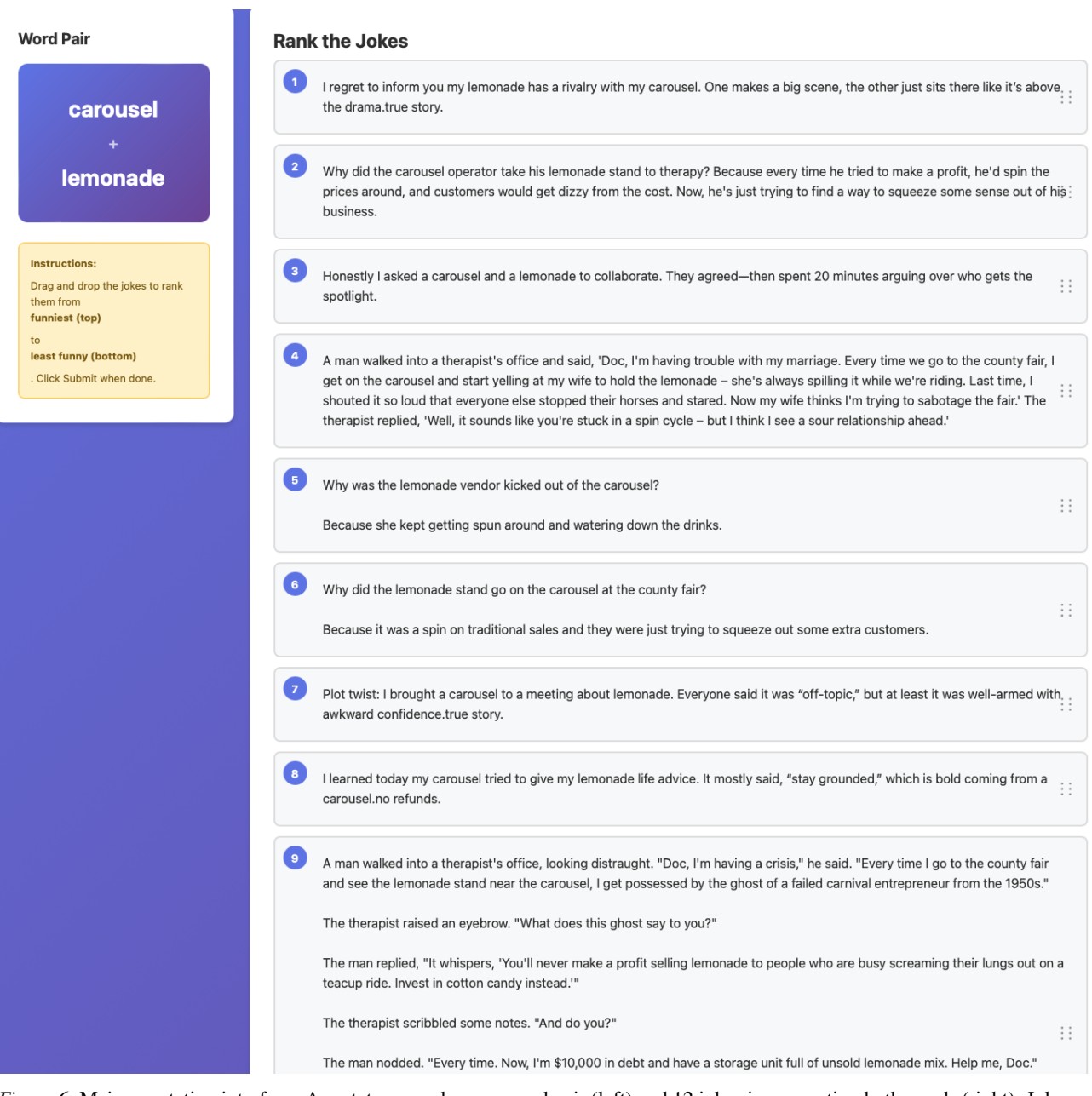

*Figure 6.* Main annotation interface. Annotators are shown a word pair (left) and 12 jokes incorporating both words (right). Jokes are ranked from funniest (top) to least funny (bottom) via drag-and-drop.

# E. Prompts

---

**Story Writing Rubric Judge**

You are given an input prompt, and a story on those lines. Compare the given story on these criteria:
• Character authenticity and insight
• Interesting and original
• Writing quality
• Coherence in plot, character choices, metaphor
• Instruction following (followed the prompt)
• World and atmosphere
• Avoids cliches in characters, dialogue & plot
• Avoids flowery verbosity & show-offy vocab maxxing
• Avoids gratuitous metaphor or poetic overload

**Judging notes:**
• Be aware that these abilities may be independent, i.e. a model may be strong in one and weak in another.
• Outputs will sometimes be truncated to ensure length consistency. Don't penalise this, just judge what is there on its merit.
• Give a score between 0 and 10 for each criterion and then give an overall score between 0 and 10 for the story.

Rate the story on a scale of 1 to 10 for each criterion. Then finally give an overall rating for the story.

```
#Input Prompt
{prompt}
#Story
{story}
```
Give your final answer in the following format:
```
<answer>5/10</answer>
```

---

**Story Writing Pairwise Judge**

Compare the relative ability of each writer on these criteria:
• Character authenticity and insight
• Interesting and original
• Writing quality
• Coherence in plot, character choices, metaphor
• Instruction following (followed the prompt)
• World and atmosphere
• Avoids cliches in characters, dialogue & plot
• Avoids flowery verbosity & show-offy vocab maxxing
• Avoids gratuitous metaphor or poetic overload

**Judging notes:**
• Be aware that these abilities may be independent, i.e. a model may be strong in one and weak in another.
• Outputs will sometimes be truncated to ensure length consistency. Don't penalise this, just judge what is there on its merit.
• You must always pick a winner for each criteria (no draws).

Rate both the writers on a scale of 1 to 10 for each criterion. Then finally give an overall rating for the writer. Based on the overall rating, pick the better writer.

```
#Input Prompt
{prompt}
#Writer A's response
{writer_a}
#Writer B's response
{writer_b}
```
Give your final answer (the winning writer) in the following format:
```
<answer>A/B</answer>
```

## Joke Pairwise Judge

You are judging which of two jokes is better for this writing task.
**Task:** Given two words, write a short, funny, coherent joke that:
• Clearly and naturally includes both words
• Is actually humorous (witty, playful, or surprising)
• Is easy to understand on first read
• Is not offensive, hateful, or needlessly cruel
```
##Input
Words:  {word1} and {word2}
##Writer A's joke
{writer_a}
##Writer B's joke
{writer_b}
```
**Evaluate the jokes on:**
• Uses both words clearly and naturally
• Funniness / wit
• Coherence and clarity
• Originality (not a generic or template-y one-liner)
• Tone (light, playful; avoid edgy/offensive content)
You must pick a single winner overall (no ties).
Respond in this format only:
```
<answer>A/B</answer>
```

## Joke Rubric Judge

You are grading a single joke for this task.
**Task:** Given two words, write a short, funny, coherent joke that:
• Clearly and naturally includes both words
• Is actually humorous (witty, playful, or surprising)
• Is easy to understand
• Avoids offensive, hateful, or needlessly cruel content
```
##Input
Words:  {word1} and {word2}
##Joke
{joke}
```
**Score the joke on these criteria (1–10 each):**
• Uses both words clearly and naturally
• Funniness / wit
• Coherence and clarity
• Originality
• Tone (light, playful; not offensive)
Then give an overall score (1–10) for the joke quality on this task.
Answer ONLY with:
```
<answer>X/10</answer>
```
where X is the overall score.

## Joke Simple Judge

You are grading a joke for a simple two-word joke task.
**Task:** Given two words, write a short, funny joke that clearly includes both words.
```
Words:  {word1} and {word2}
Joke:
{joke}
```
On a scale from 1 to 10, how good is this joke for the task? Consider only:
• Does it include both words clearly?
• Is it at least mildly funny / playful?
• Is it coherent and understandable?
Answer ONLY with:
```
<answer>X/10</answer>
```

