# OpenReview forum: "Tournament Style RL: Stabilizing Policy Optimization on Non Verifiable Problems"
_ICML.cc/2026/Conference — ICML 2026 regular_

### Official Review · Reviewer_b8du · 2026-03-01

**Soundness:** 3
**Presentation:** 3
**Significance:** 3
**Originality:** 2
**Overall Recommendation:** 4
**Confidence:** 3

**Summary:**

This paper aims to address the challenge of applying reinforcement learning to non-verifiable problems. The authors proposed a method called TSRL. It first prompts other LLMs to generate a fixed set of reference anchors. During training, the model’s generated responses are compared against these anchors one by one, and the win-rate is used as the signal for optimization. The authors evaluate this approach across four non-verifiable problems, showing that TSRL significantly outperforms the baseline methods and enables a more robust learning process.

**Compliance With Llm Reviewing Policy:**

Affirmed.

**Final Justification:**

The supplementary experiments illustrate the efficiency of this method. Although I still feel this work does not solve the fundamental problem in unverifiable RL, it provides a solution from the engineering side. The paper is overall more solid after rebuttal.

**Key Questions For Authors:**

See weaknesses.

**Limitations:**

Yes.

**Strengths And Weaknesses:**

**Strengths**

- Reinforcement learning for non-verifiable problems is currently a widely discussed topic. The authors are attempting to solve an important and relevant issue.
- The experiments are comprehensive. The authors tested their method on four subjective tasks, where TSRL demonstrated consistent performance improvements. Furthermore, the authors provide rich ablation studies and sensitivity analyses for hyperparameters.
- Objective and fair evaluation. In addition to LLM-as-a-Judge evaluation, the authors also conducted a blinded human evaluation, which makes the results much more convincing.

**Weaknesses**

- The names of the baselines in Lines 275-294 are inconsistent with those in Table 1 and Figure 2. The training curve for the intrinsic baseline is missing from Figure 2.
- During training, TSRL requires an LLM judge to compare each sampled completion with the anchors, so the overall overhead is likely dominated by the judge's inference cost. However, the paper does not systematically report these costs or provide a comparison with the baselines, making it difficult to evaluate whether the method can scale.
- In Line 303, the authors mention setting the training steps for several baselines to 4x that of TSRL to account for the additional Judge calls used in TSRL, and further claim that TSRL is compute efficient because it uses fewer gradient steps. I do not think this is a strictly fair alignment. The communication overhead and inference FLOPs required to call external large models for pairwise comparisons are nonnegligible. Claiming that it is compute efficient merely because of fewer backpropagation steps is an overclaim.
- To distinguish whether the performance improvement comes from the stable reference frame provided by the fixed anchors or from the noise reduction of multiple comparisons, I suggest adding a baseline of pairwise comparisons among the 8 completions sampled within a single batch.

---

> ### Author Rebuttal · Authors · 2026-03-30
>
> We thank the reviewer for the thorough and constructive review, and for recognizing the comprehensive experiments, rich ablations, and the value of blinded human evaluation. Here are our clarifications to the questions raised:
>
> ---
>
> **> W1: "Baseline names inconsistent between Lines 275-294 and Table 1/Figure 2. Intrinsic baseline training curve missing."**
>
> Thank you for pointing this out. We will fix the baseline names for consistency and add the training curve for the intrinsic baseline in Figure 2 in the camera-ready version.
>
> ---
>
> **> W2: "Judge inference cost not reported. Difficult to evaluate scalability."**
>
> The total judge calls for each method are:
>
> - Rubric-based / Single-comparison baselines: batch_size x GRPO_group_size x num_gradient_steps
> - TSRL: batch_size × GRPO_group_size × num_gradient_steps × k (num_anchors)
>
> TSRL uses k times more judge calls per gradient step. To account for this, we run the baselines for k times more gradient steps (Line 303), equalizing total judge calls. This comes to about `$21.12` each for TSRL and rubric judge baseline during training and `$3.96` TSRL and each baseline for inference. Under this equal-budget comparison, TSRL still outperforms all baselines.
>
> We also ran all methods at a fixed 220 gradient steps (without the 4x multiplier for baselines). TSRL is more expensive in total judge calls, but the performance gain more than compensates:
>
> | Method | 220 steps | 880 steps |
> |--------|:---------:|:---------:|
> | Base Model | 13.5% | 13.5% |
> | RLHF (RM) | 37.5% | 40.5% |
> | Rubric-based | 41.5% | 47.5% |
> | TSRL | **74.0%** | — |
>
> TSRL at 220 steps outperforms baselines at 880 steps, making it Pareto-optimal on the performance-vs-judge-cost curve. We will add a performance vs. total judge cost plot over all tasks and models in the camera-ready version.
>
> ---
>
> **> W3: "Claiming compute efficiency from fewer gradient steps is an overclaim."**
>
> We agree that framing efficiency purely in terms of gradient steps is incomplete and will revise this claim. The more precise statement is: at equal total judge inference budget, TSRL achieves higher final performance. As shown above, baselines given 4x the gradient steps (and thus equal judge calls) still underperform TSRL. If we assume task performance improves with continued training, matching TSRL's performance would require even more gradient steps and substantially more judge calls. TSRL is more efficient per unit of judge compute, not per gradient step alone. We will rewrite Line 303 accordingly.
>
> As a side note, TSRL on stories task for LLaMA 3.1 8B takes **2.5 hours** to train compared to **~7 hours** for rubric baseline. We will also add a performance v/s train time comparison plot in the camera ready version.
>
> ---
>
> **> W4: "Add a baseline of pairwise comparisons among the 8 in-batch completions to disentangle stable reference frame vs. noise reduction from multiple comparisons."**
>
> Excellent suggestion. We ran this on the story generation task, comparing all 8 sampled completions against each other within each batch and assigning a reward equal to the in-batch rank (e.g., the 3rd-best completion gets reward 6/8, with advantage computed relative to the group mean of 9/16).
>
> | Method | Win Rate (Stories) |
> |--------|:------------------:|
> | In-batch pairwise ranking | 55.0% |
> | TSRL (k=8 fixed anchors) | **74.0%** |
>
> These results confirm that the stable reference frame is the key driver of improvement, along with variance reduction of course. In-batch ranking still suffers from a shifting baseline. We will add the results on all tasks in the camera ready version.
>
> ---
>
> We hope these responses address the points raised and kindly invite the reviewer to consider increasing their score accordingly.

---

> > ### Author Rebuttal · Reviewer_b8du · 2026-04-03
> >
> > Thanks for the clarification which well solved my questions about efficiency. I raised my score accordingly.

---

### Official Review · Reviewer_yDNS · 2026-03-12

**Soundness:** 2
**Presentation:** 3
**Significance:** 2
**Originality:** 2
**Overall Recommendation:** 4
**Confidence:** 3

**Summary:**

In this work, the authors proposed Tournament Style Reinforcement Learning, which replaces the scalar rewards with a win-rate signal obtained by rubric-guided pairwise judgments against a fixed set of anchor responses. The experiments were conducted with four tasks and two LLMs, and the results indicate that TSRL improves the average win-rate over the base models. Then, the authors verified the gains of using TSRL with a human preference study with 6 participants.

**Compliance With Llm Reviewing Policy:**

Affirmed.

**Ethical Review Concerns:**

I did not use AI for reviewing, but I did use AI tools like GPTZero to verify, since many parts of this paper seem very suspicious. At least for the parts I examined (abstract, intro, first paragraph in the appendix), they are all flagged as AI-generated.

**Ethical Review Flag:**

Flag this paper for an ethics review.

**Ethics Expertise Needed:**

["Research Integrity Issues (e.g., plagiarism)"]

**Final Justification:**

Most of my concerns have been addressed. The paper and the results seem promising to me.

**Key Questions For Authors:**

1. How was LLM used in paper writing? Are equations also generated by AI?
2. Could you provide more insignts about how this method would generalize to other RL methods beyond LLMs?

**Limitations:**

Some limitations were mentioned.

**Strengths And Weaknesses:**

--------------------------------Strength----------------------------------------------------------
1. The idea of the paper looks promising
2. The terms are well-defined
3. I appreciate the verification with real humans.

--------------------------------Weakness-------------------------------------
1. My largest concern is that a lot of content seems to be generated by AI. I would ask the authors to kindly share their AI usages during the paper writing to reduce my personal concerns about the validity of the results.
2. A lot of contributions are claimed without any evidence as support. For example, "TSRL can be also applied to robotics and embodied tasks where success is often not cleanly verifiable and reward functions are difficult to specify (e.g., towel folding, table setting, or other deformable-object manipulation).", while the authors only tested the performance of the method in training LLMs with distilled-ish rewards.
3. some details are missing. for instance: what RLHF method was used as baseline? Please list citations.

Overall, I would recommend rejecting the paper due to unclear LLM usage. However, the paper targets an interesting problem and the results seem convincing. It would push the paper into the acceptance territory if the authors are willing to provide LLM usage explanation, and drop some over-claimed contributions,

---

> ### Author Rebuttal · Authors · 2026-03-30
>
> We thank the reviewer for their constructive feedback. We appreciate the reviewer for recognizing that TSRL is promising. Here are our responses to the clarifications requested.
>
> ---
> **>W1: "A lot of content seems to be generated by AI. Share AI usage."**
>
> In compliance with the ICML guidelines, we acknowledge that we use LLMs to refine the prose and improve the grammatical clarity of certain segments of the paper. We however emphasize that all core ideas, experimental designs, and research artifacts presented in this work are developed entirely by human authors.
> To provide a more details, we use ChatGPT (the $20 version) to refine the writing. For example, we provide bullet-point notes about our human study (setup, annotator count, agreement scores, win rates) and ask ChatGPT to "write it in a better style." The LLM returns a polished LaTeX paragraph which we then further refine. All the scientific content, what to measure, how to set up the study, what numbers to report, all are hand written by the authors. For the camera-ready version, we will add an appendix section detailing where and how LLMs were used.
>
> ---
> **>W2: "Contributions claimed without evidence. Robotics/embodied tasks mentioned but only LLMs tested."**
>
> Thanks for pointing this out. We state this line in the future work section. We do not claim in this paper that TSRL can be used right away for robotics tasks. The intended message was that,  with some modification, a method like TSRL (or motivated by TSRL) could potentially be applied to robotics, since there are many non-verifiable tasks for which reward design is difficult. For the camera-ready version, we can drop this paragraph to avoid any ambiguity.
>
> ---
> **>W3: "What RLHF method was used as baseline? Missing citations."**
>
> We follow the standard setup of Ouyang et al. (2022) [1], train a reward model(LLaMA-3.1-8B) on pairwise preferences between the same anchors used for TSRL, then optimize the policy (LLM being trained) using that reward model. We will add these citations in the camera ready version.
>
> ---
> **>Q1: "How was LLM used in paper writing? Are equations also generated by AI?"**
>
> Thanks for pointing this out. We would like to clarify that, No, the equations are not AI-generated. These are standard equations in RL (policy gradient estimator, variance of gradient estimator) and probability (Hoeffding's inequality, Bernoulli assumptions, expected value and variance). Everything else is derived on top of these. AI is only used for grammatical refinement of the language, as shown in the example above.
>
> ---
> **>Q2: "How would this method generalize to other RL methods beyond LLMs?"**
>
> Thanks for pointing this out.  In the scope of this paper, we do not claim that TSRL in its current form applies directly to any robotics setting. That said, we do believe that with VLMs used to judge the quality of a robotic trajectory, compared against baseline trajectories sampled or collected via teleoperation, a robot could be trained to perform better on certain non-verifiable tasks. For example to train a robot to clean a room, there could be multiple trajectories to tidy up the room. The anchors here are a ranked set of diverse trajectories to achieve the outcome - clean room.
>
> But, again, this remains a potential and interesting future direction, not a claim of this paper.
>
> ---
> **Regarding the Ethics Flag**
>
> The reviewer mentions using GPTZero to flag portions of the paper as AI-generated. As indicated above, we use AI for language polishing only, which is permitted under ICML policy. We do not believe a GPTZero flag constitutes evidence of research integrity concerns. All ideas, experiments, and artifacts were obtained through rigorous manual effort. We will release code, data, and experiment logs upon acceptance and are happy to provide them during the rebuttal phase. We respectfully note that automated AI-detection tools have high false-positive rates on technical academic writing [2,3,4], making them unreliable for authorship judgments.
>
> ---
> We thank the reviewer for their thoughtful feedback. We hope our clarifications address the concerns and respectfully request the reviewer to consider a score increase.
>
> [1] Training language models to follow instructions with human feedback, Ouyang et.al. 2022.
>
> [2] Policies Permitting LLM Use for Polishing Peer Reviews Are Currently Not Enforceable, Saha et al. 2026
>
> [3] Almost AI, almost human: The challenge of detecting AI-polished writing, Saha et al. 2025
>
> [4] LLM-as-a-coauthor: Can mixed human-written and machine- generated text be detected?, Zhang et al. 2024

---

> > ### Author Rebuttal · Reviewer_yDNS · 2026-04-03
> >
> > I appreciate the author's response.
> >
> > The paper addressed an interesting problem and used a different way to extract rewards from VLMs. The application seems promising, while the contributions are slightly on the incremental side.
> >
> > I would strongly encourage the authors to include an LLM usage statement in the paper and a more detailed statement into appendix. While I am still curious about the reproducibility since the author did not open-source their code, I am willing to raise my score.

---

> > > ### Author Response · Authors · 2026-04-06
> > >
> > > We thank the reviewer for their evaluation and willingness to raise the score.
> > >
> > > **Code Release.** We anonymously release the codebase for this project, including data curation, training pipeline, evaluation scripts, and a README with reproduction instructions, available **[*here*](https://anonymous.4open.science/r/tsrl-anonymous-20260406-54FD/README.md)**. We hope this addresses the reproducibility concern.
> > >
> > > **LLM Usage Statement.** We will add the following statement to the paper: *In compliance with ICML guidelines, we acknowledge the use of LLMs to refine prose and improve grammatical clarity in certain segments of the paper. Specifically, we use ChatGPT to polish writing given hand-written bullet-point notes. All core ideas, experimental designs, research artifacts, and scientific content are developed entirely by the human authors.*
> > >
> > >  We hope that our response addresses the reviewer's concern. If so, we kindly ask the reviewer to consider revising their score.

---

### Official Review · Reviewer_k6Vx · 2026-03-16

**Soundness:** 3
**Presentation:** 2
**Significance:** 2
**Originality:** 3
**Overall Recommendation:** 4
**Confidence:** 3

**Summary:**

The paper proposes a novel tournament reward calculated against a given set of anchor answers to generate reward supervision for LLM training in tasks without verifiable rewards. For each input prompt, a set of anchors is generated before training using a stronger LLM and ranked. Then the generated answers are compared against this ranked set of anchors to generate a reward for each answer which is then used for GRPO fine-tuning.

**Compliance With Llm Reviewing Policy:**

Affirmed.

**Final Justification:**

My only remaining concern is that the ranking reward system, at the end, still relies on distilling knowledge from another bigger model. It does not solve the root question "where do we even get the very first reward automatically for non-verifiable problems?" This is quite an open question to which we don't have good answers other than distilling another model. The proposed ranking reward is a nice contribution but still doesn't answer this fundamental question so I'll preserve my current rating (still positive though).

The newly added experiments in the reply partially mitigate my concern, but I am worried that bootstrapping the exact model under for reward would lead to more severe reward hacking. Intuitively, I would expect a solution directly extracted from human preferences or some learned rubrics from human annotations.

**Key Questions For Authors:**

1. Anchor quality is vital to the success of the proposed method. Could you provide more evaluations on anchors generated by different models and how the quality of anchors affect fine-tuning performance? The quality of anchors could be either evaluated by another model or even better, by human.
2. Can we train reward models using the generated score then fine tune with trained reward model? Will that lead to the same good results with better scalability?

**Strengths And Weaknesses:**

**Strength**
1. the proposed method is well motivated and clearly presented
2. can be easily implemented upon GRPO style fine-tuning pipelines
3. is robust against noise in evaluator LLMs

**Weakness**
1. The proposed method relies heavily on anchor model quality. And the score itself will saturate if the model being fine-tuned surpasses the anchor model's quality. On the other hand, the performance of anchor model upper limits the model been fine-tuned.
2. Anchor model generation diversity is also crucial to the success of this reward. But currently we don't have very straightforward & easy control for arbitrary generation diversity while maintaining quality.
3. It would be interesting to see if the proposed method can beat or provide on-par performance against existing reward models in fully verifiable or partially verifiable tasks. Also, no comparison against DPO is provided.

---

> ### Author Rebuttal · Authors · 2026-03-30
>
> We thank the reviewer for the positive feedback and recognizing that TSRL is well motivated, easy to implement, and robust to judge noise. Here are the clarifications and additional experiments as requested:
>
> ---
>
> **> W1: "The proposed method relies heavily on anchor model quality. The score will saturate if the fine-tuned model surpasses the anchors."**
>
> Yes, we agree with the reviewer that as the model improves, the anchors would get saturated. That's why we suggest the following in the future work section:
>
> "... Refresh anchors over training: periodically re-sample a diverse anchor pool from the improved policy (or a mixture of policy snapshots) to maintain a discriminative reference ladder. This would turn TSRL into a self-improving procedure that reduces reliance on external anchor sources, and could ultimately enable training from model-generated reference sets once a sufficiently strong initial policy is available. …"
>
> This is an interesting and immediate future direction but out of scope for the current paper. It could also be viewed as an extension of how Kimi K2 does self-critique policy optimization [1]. The central claim of this work is that a stable reference frame provides a signal more robust to judge noise compared to existing single-comparison approaches.
>
> ---
>
> **> W2 and Q1: "Anchor model generation diversity is crucial but hard to control while maintaining quality."**
>
> To ensure diverse, high-quality anchors, we sample from a larger LM (Gemini-2.5-Pro) at different temperatures and with different system prompts. We discuss importance of diversity in Section 4.5 by constructing high-diversity and low-diversity anchors at the same k.
>
> To show the effect of anchor quality, we generate anchors with Gemini-2.5-Pro and Gemini-2.5-Flash and train LLaMA-3.1-8B on story generation:
> | Method | Win Rate |
> |--------|:--------:|
> | Base model | 13.5% |
> | Rubrics (strongest baseline) | 47.5% |
> | TSRL (human-written stories) | **74.0%** |
> | TSRL (Gemini 2.5 Pro) | **70.0%** |
> | TSRL (Gemini 2.5 Flash) | 65.5% |
>
> We also perform a qualitative study using GPT-5.4 as judge. For each story, we sample 4 anchors from both models and compare them pairwise in order of ranking (4 judge calls per story). Gemini 2.5 Pro anchors beat Gemini 2.5 Flash anchors in 81.2% of cases. The model trained with Pro anchors beats the Flash-anchored model in 54.5% of cases. We will report results on all tasks in the camera-ready version.
>
> ---
>
> **> W3: "Comparison with reward models on verifiable tasks and with DPO."**
>
> For fully verifiable domains where correctness depends only on the final output, creating a ranking between responses (as TSRL does) is not practical. But for partially verifiable domains (like CoT in math), we can rank solutions by degree of correctness. For example, a solution with a late mistake ranks higher than one with an early mistake..
>
> We compare TSRL to binary (final correctness) reward using GRPO on MATH. We prompt Gemini-2.5-Pro to generate anchors: a) completely incorrect, b) mistakes in the first half, c) mistakes in the second half, d) gold solution. We finetune LLaMA-3.1-8B for 100 gradient steps using Kimi-K2 as judge. **Base CoT: 47.0%; TSRL: 54.5%; Binary reward: 50.0%.**
>
> **Comparison with DPO.** We finetune using DPO where pairs are the generated anchors on the stories dataset. Results for LLaMA-3.1-8B:
>
> | Method | Win Rate |
> |--------|:--------:|
> | DPO | 57.5% |
> | TSRL | **74.0%** |
>
> We will add full tables for all tasks on both LLaMA and Qwen models in the camera-ready version. We will also add comparisons against outcome-based rewards and process-based rewards (PRM).
>
> ---
>
> **> Q2: "Can we train a reward model from the tournament scores, then fine-tune with that reward model for better scalability?"**
>
> Yes, we explored this with the RLHF(RM) baseline where we created preference pairs from anchor responses to train a reward model. We then used the trained RM to finetune the LLM. TSRL still outperforms this technique as shown in Table 1.
>
> ---
>
> Thank you again for the constructive review. We believe the additional experiments and clarifications have strengthened the manuscript. We remain open to further discussion and, if your concerns have been resolved, we hope you will consider updating your evaluation.
>
> [1] Kimi K2: Open Agentic Intelligence

---

> > ### Author Rebuttal · Reviewer_k6Vx · 2026-03-31
> >
> > Thanks for the clarifications and the additional experiment results provided. My only remaining concern is that the ranking reward system, at the end, still relies on distilling knowledge from another bigger model. It does not solve the root question "where do we even get the very first reward automatically for non-verifiable problems?" This is quite an open question to which we don't have good answers other than distilling another model. The proposed ranking reward is a nice contribution but still doesn't answer this fundamental question so I'll preserve my current rating (still positive though).

---

> > > ### Author Response · Authors · 2026-04-06
> > >
> > > We thank the reviewer for their continued engagement and for raising the important question of **obtaining ranking rewards automatically**, without reliance on a stronger external model.
> > >
> > > **Self-Bootstrapped TSRL**. We test a variant where **anchors are sampled from the policy being trained**, eliminating any dependence on a larger model.
> > >
> > > We test on story generation task, finetuning Llama-3.1-8B-Instruct. Training hyperparameters match the main experiments. lr=5e-7, batch size 8, rollout n=16, KL penalty 0.001, 8×A100 GPUs.
> > >
> > > **Self-bootstrapped anchor generation.** Instead of sourcing anchors from a larger model or human writers, we sample 4 anchor stories per train prompt from the policy(LLM) being trained, using system prompts that explicitly target different quality tiers:
> > >   - Tier 1 (low-effort): "You are a student... "
> > >   - Tier 2 (mediocre): "You are an average hobbyist writer... aim for 'okay', not 'good'."
> > >   - Tier 3 (skilled): "You are a skilled professional fiction writer... aim for a solid published magazine piece."
> > >   - Tier 4 (top-tier): "You are a top-tier, award-winning literary fiction writer... every sentence should do work."
> > >
> > > We train for 200 steps, and at every 50 steps, we re-sample the anchors from the current policy. We evaluate at every checkpoint (after every 50 steps) using held-out test set with human-generated anchors identical to the main experiments.
> > >
> > > Results:
> > >
> > > | Method | Step | Win Rate | Δ from baseline|
> > > |---|---|---|---|
> > > | Base model (no training) | 0 | 13.5 | — |
> > > | Iter 1 Self Bootstrap TSRL | 50 | 40.5 | +27.0 |
> > > | Iter 2 Self bootstrap TSRL | 100 | 77.5 | +64.0 |
> > > | Iter 3 Self bootstrap TSRL | 150 | 80.0 | +67.5 |
> > > | Iter 4  Self bootstrap TSRL | 200 | 81.5 | +69.0 |
> > > | TSRL | 220| 74.0 |+60.5|
> > > |Rubrics |880|47.5|+34.0|
> > >
> > >
> > > ---
> > >
> > > Self-Bootstrap TSRL achieves **81.5% win rate**, a **+8.5 point improvement** over standard TSRL and **+34 points** over rubric-based reward with no reliance on a stronger model. We hypothesize that periodically re-sampling anchors from the training policy creates a curriculum yielding orthogonal gains on top of TSRL’s stable reference frame.
> > >
> > > We again thank the reviewer for asking the question. We hope this addresses the reviewer's concern. If so, we kindly ask the reviewer to consider revising their score.

---

### Official Review · Reviewer_Lqn1 · 2026-03-20

**Soundness:** 3
**Presentation:** 4
**Significance:** 3
**Originality:** 3
**Overall Recommendation:** 4
**Confidence:** 2

**Summary:**

Non-verifiable problems are common in real-world settings, and designing reward functions for such problems is very challenging. The paper aims to address non-verifiable problems. It introduces Tournament-Style Reinforcement Learning (TSRL), which constructs rewards from rubric-guided pairwise judgments against a fixed set of anchor responses and uses the win rate as the reward for policy optimization. Experiments on four non-verifiable tasks and two backbone LLMs demonstrate the efficiency of the proposed method.

**Compliance With Llm Reviewing Policy:**

Affirmed.

**Key Questions For Authors:**

It is not clear whether the method still uses GRPO for policy learning.

What do “relative pairwise judgments” mean?

What is meant by a fixed set of anchor responses, and how are these responses generated?

How is the size of the anchor response set determined?

What does “win rate” refer to?

How to ensure the quality of anchors and define the size of the anchor set? There might be some analysis or experiments.

**Limitations:**

It would be better to provide more details about how to generate the fixed set of anchor responses.

**Strengths And Weaknesses:**

Strengths:

Scaling RL to non-verifiable problems remains an unsolved and interesting challenge.

The idea of using a fixed set of anchor responses is novel and provides a more robust signal.

The experimental results demonstrate better performance of the proposed method.

Weaknesses:

It would be better to provide more details about anchor construction. Like how to generate anchors and how to ensure their qualities.

The motivation using GRPO is not clear.  Can the approach use standard policy methods?

Experiments are limited. It would be better to use more models. The paper does not show results on different sizes of anchor set.

---

> ### Author Rebuttal · Authors · 2026-03-30
>
> We sincerely thank the reviewer for their thoughtful and positive review. Here are the clarifications and additional results as requested:
>
> **> W1: "More details about anchor construction. How to generate anchors and ensure their quality."**
>
> We describe anchor construction in Section 3 (lines 261-274). The construction depends on availability:
>
> - For **Story generation**, we use the WritingPrompts dataset that already had ~10 human-written completions per prompt (out of which we select 4 well-separated responses, after the tournament, as anchors).
> - For **other tasks**, we prompt Gemini 2.5 Pro at varied temperatures and diverse quality prompts to generate responses spanning a quality spectrum.
>
> We rank anchors using rubric-guided pairwise comparisons followed by topological sort to produce a global ordering. We ensure quality and diversity by prompting strong generators (humans or frontier LLMs) with diverse instructions as explained in Section 4.3. Table 1 shows that anchor diversity is an important design factor in TSRL. We will add the prompts in the appendix in the camera-ready version.
>
> **> W2: Motivation of using GRPO and use of other policy gradient methods.**
>
> TSRL is a **reward design mechanism** rather than a policy optimization method. It can be used with any policy gradient method. The theory behind noise stability and variance reduction uses the standard REINFORCE gradient estimator (Section 2.2). We chose GRPO because it is the current SOTA for RL in LLMs, but the same reward would work for PPO, REINFORCE, or any on-policy RL algorithm. We will add a comparison table with PPO in the appendix of the camera-ready version.
>
> **> W3: Experiments are limited. Results on more models. Results on different anchor sizes.**
>
> We evaluate TSRL on two backbone LLMs (LLaMA-3.1-8B and Qwen-3-4B) across 4 tasks. We further train Qwen2.5-3B-Instruct and Qwen2.5-7B-Instruct using TSRL (up to 8B due to resources) and compare with Rubric-based reward optimization (best baseline) on story generation:
>
> | Method | Qwen2.5-3B-Instruct | Qwen2.5-7B-Instruct |
> |--------|:-------------------:|:-------------------:|
> | Base   | 10.0%               | 15.3%               |
> | Rubric | 45.0%               | 50.0%               |
> | TSRL   | **59.2%**            | **78.0%**            |
>
> Full results on all tasks will be added in the final paper.
>
> Regarding anchor set sizes, Figure 3(a) reports this ablation. We sweep k ∈ {1,2,4,5,6,8,10} across all four tasks and both models, finding large gains from k=1 to k=4 and diminishing returns beyond k=8.
>
> **> Q1: Does the method still use GRPO?**
>
> Yes. However, TSRL just provides a reward function, completely decoupled from the optimization algorithm (also explained in W2).
>
> **> Q2: What do relative pairwise judgements mean?**
>
> Given two responses *a* and *b*, the LLM judge outputs a binary decision: "*a* is better" or "*b* is better", guided by a task-specific rubric (the same rubric used for the rubric baseline, see Appendix D). This is *pairwise* because it compares a pair of responses, and *relative* because it compares sampled responses to get a reward as opposed to absolute scoring.
>
> **> Q3: What is a fixed set of anchor responses and how are they generated?**
>
> We Discuss this in Section 3, Lines 262–273. For each prompt *x*, we generate *k* responses {r₁,...,rₖ} before training, these are called anchors and are written by humans or generated via an LLM. These "anchors" are fixed throughout training — the policy gets better, but the anchors remain the same, providing a stable reference frame.
>
> What happens when the LLM surpasses all anchors? We leave this as future work, with one solution being to periodically re-sample anchors from the improved policy (discussed in Future Work).
>
> **> Q4: How is the size of anchor response set determined?**
>
> Discussed in Section 4.2 and Figure 3(a). Performance scales with *k*, but we find diminishing returns past k=4; from k=8–10 we observe marginal improvement. Hence for the main results we use k=4 anchors.
>
> **> Q5: What does win rate refer to?**
>
> Defined in Section 2.2. For a candidate response *c*, w(c) is the fraction of anchors that *c* beats in pairwise comparison. If k=10 and *c* beats 7, w(c)=0.7. This is the reward function for RL during training.
>
> **> Q6: Analysis and experiment to determine the effect of quality and size of anchor set.**
>
> Effect of size is discussed in Section 4.2 and Q4. Effect of quality and diversity is discussed in Section 4.3 and W1. We are happy to run any more experiments if needed.
>
> ---
>
> We reiterate our sincere appreciation to the reviewer. We are happy to answer any more questions or run more experiments to establish the utility of TSRL. If the reviewer finds our responses satisfactory, we kindly request them to consider revising their score.

---

### Decision · Program_Chairs · 2026-04-30

**Decision:**

Accept (regular)

**Comment:**

The authors provide a novel method called Tournament-Style Reinforcement Learning to address the problem of reward design in non-verifiable problems.

The paper solves an important challenge, and the idea of using anchor responses is novel.

Some reviewers found that the experimental evaluation is limited, but the authors provide new experiments with new baselines and ablations during the rebuttal. One reviewer pointed out the current challenge of how to obtain ranking rewards automatically without an external model. This challenge is not fully solved by the authors, and I suggest that they add a discussion about it in the paper.